# Structure of an Intranucleosomal DNA Loop That Senses DNA Damage during Transcription

**DOI:** 10.3390/cells11172678

**Published:** 2022-08-28

**Authors:** Nadezhda S. Gerasimova, Olesya I. Volokh, Nikolay A. Pestov, Grigory A. Armeev, Mikhail P. Kirpichnikov, Alexey K. Shaytan, Olga S. Sokolova, Vasily M. Studitsky

**Affiliations:** 1Biology Faculty, Lomonosov Moscow State University, 119992 Moscow, Russia; 2Department of Pharmacology, Rutgers Robert Wood Johnson Medical School, Piscataway, NJ 08854, USA; 3Shemyakin-Ovchinnikov Institute of Bioorganic Chemistry, Russian Academy of Sciences, 117997 Moscow, Russia; 4Faculty of Biology, MSU-BIT Shenzhen University No. 1, International University Park Road, Dayun New Town, Longgang District, Shenzhen 518172, China; 5Fox Chase Cancer Center, Philadelphia, PA 19111, USA

**Keywords:** nucleosome, transcription, RNA polymerase II, elongation complex, structure, DNA damage, molecular modeling

## Abstract

Transcription through chromatin by RNA polymerase II (Pol II) is accompanied by the formation of small intranucleosomal DNA loops containing the enzyme (i-loops) that are involved in survival of core histones on the DNA and arrest of Pol II during the transcription of damaged DNA. However, the structures of i-loops have not been determined. Here, the structures of the intermediates formed during transcription through a nucleosome containing intact or damaged DNA were studied using biochemical approaches and electron microscopy. After RNA polymerase reaches position +24 from the nucleosomal boundary, the enzyme can backtrack to position +20, where DNA behind the enzyme recoils on the surface of the histone octamer, forming an i-loop that locks Pol II in the arrested state. Since the i-loop is formed more efficiently in the presence of SSBs positioned behind the transcribing enzyme, the loop could play a role in the transcription-coupled repair of DNA damage hidden in the chromatin structure.

## 1. Introduction

Single-strand breaks (SSBs, nicks) are among the most common DNA lesions, representing nearly 75% of all DNA damages arising daily in a mammalian cell [1]. They are generated in amounts from tens to hundreds of thousands during various processes of DNA metabolism (topoisomerase action, DNA repair by base or nucleotide excision pathways, etc.) and after exposure to environmental and intracellular genotoxins [1,2,3,4]. Unrepaired SSBs can interfere with different DNA transactions (transcription, replication and repair), induce the accumulation of double-stranded DNA breaks, increase genomic instability and lead to apoptosis and cancer; the dysfunction of proteins involved in SSB repair (SSBR) causes severe hereditary diseases (reviewed in [1,4,5,6,7]). Recent research has revealed a role of the SSBR system as a potential target for the regulation of cellular senescence and as a key player in a maintenance of neuronal integrity and plasticity [8,9,10,11,12]. Therefore, neural tissue is more sensitive to SSBR dysfunction, leading to various neurological disorders, such as age-associated and hereditary neurodegenerative diseases [4,5,13,14].

In most cases, 5′- and/or 3′-ends of an SSB are modified and, therefore, are not available for immediate ligation (described in [3,4]). Usually, SSBs are recognized by AP endonuclease 1 (APE1) [15], or by poly [ADP-ribose] polymerase enzymes PARP1 and PARP2 and repaired through the SSBR pathway [16,17,18,19,20,21]. In some cases, nicks are removed via homologue-mediated recombination or repair pathways (homology-directed repair, HDR) [22]. However, in eukaryotic cells, DNA is organized in chromatin, which considerably limits DNA-binding proteins’ access to DNA and interferes with the recognition of at least some SSBs [23]; chromatin remodelers promote SSBR in nucleosomes [24,25,26]. Some otherwise undetectable SSBs can be sensed by processive enzymes progressing along DNA. In particular, SSBs localized on template DNA strand block the progression of RNA polymerase II (Pol II) in vitro and in vivo [27,28]. Stalled Pol II serves as a signal to initiate the process of transcription-coupled nucleotide excision repair (TC-NER); thus, in transcriptionally active cells, Pol II-blocking SSBs can be repaired through this pathway [29]. However, SSBs localized on non-template DNA strand (NT-SSBs) do not considerably affect the transcription of histone-free DNA in vitro [30], although they are efficiently repaired in vivo [31], suggesting that chromatin structure could play a role in the repair of non-template DNA strands.

Pol II transcribes the majority of eukaryotic protein-coding genes and is involved in TC-NER as a sensor. Pol II transcription induces chromatin remodeling, histone exchange and covalent modifications [32,33,34]. Although chromatin is unfolded upon gene activation in vivo, the DNA remains organized into nucleosomes in the coding region of transcribed genes [35]; histones are only transiently evicted when transcription levels are high [36,37,38]. The transcription of genes by Pol II at a moderate level is also accompanied by the retention of nucleosomal organization. Thus, typically Pol II meets nucleosomes during elongation trough every ∼200-bp of DNA. Pol II encounters a high nucleosomal barrier [39,40,41,42] and transcription through chromatin is accompanied by the preferential loss of H2A/H2B dimer both in vivo and in vitro [40,43].

Transcription through nucleosomes in vitro is accompanied by the formation of intranucleosomal DNA loops (i-loops) that could be involved in Pol II arrest during the transcription of damaged DNA and nucleosome survival during transcription [44,45,46]. Two types of the i-loops that are formed at different positions of Pol II in a nucleosome were described [44,45]. The key intermediate involved in nucleosome survival during transcription is a small i-loop (Ø-loop), formed when Pol II reaches position +49 (49 bp from the promoter-proximal nucleosome boundary) [44]. Data obtained by biochemical approaches also suggest the formation of transient intermediates containing slightly larger i-loops that are formed when the active center of Pol II reaches positions +24, +34 or +44 bp from the nucleosome boundary [45]. These i-loops have an expected arm length of about 15–20 bp and are involved in the arrest of Pol II during the transcription of DNA containing SSBs [45]. However, the existence of the i-loops has not been confirmed by structural approaches.

Here, we describe a structural analysis of the elongation complex formed when the active center of RNAP reaches position +24 bp from the nucleosome boundary, where the formation of an i-loop is expected [45]. The formation of an i-loop of the expected size, containing an enzyme backtracked by ~2–4 bp and locked in the arrested state, was detected using biochemical approaches and electron microscopy. A single SSB in non-template strand at the position +12 bp from the nucleosome boundary strongly stabilizes the i-loop and inhibits further transcription, suggesting a possible role of i-loops in transcription-coupled DNA repair.

## 2. Materials and Methods

### 2.1. DNA Templates

A plasmid-containing nucleosome positioning sequence 603 [47] was provided by Dr. Widom. The 603 sequence was modified at ten positions to construct the 603-25A+12nick template containing a single site for nicking endonuclease Nt.BsmA1 (NEB, Ipswich, MA, USA) to produce DNA break in non-template strands after the +12 position (NT-SSB) of the nucleosome and allowing for the RNAP stalling of *E. coli* at the position +24 [45]. To prepare the templates for *E. coli* RNAP transcription, the nucleosome positioning sequence digested by TspRI (NEB, Ipswich, MA, USA) was ligated through the TspRI site to the T7A1 promoter-bearing fragment [48]. The ligated product was cloned in *E. coli* cells and reamplified with 5′ end fluorescently or radioactively labeled primers and gel-purified. The following T7A1-s603-25A+12nick template was obtained: 5′CCGGGATCCAGATCCCGAAAATTTATCAAAAAGAGTATTGACTTAAAGTCTAA CCTATAGGATACTTACAGCC**ATC**GAGAGGGACACGGCGAAAAGCCAACCCAAG CGACACC*GGCACTGGGG*CCCGGTGTCTCC^GCCCGCCTGCC***G***AGTGAAATCGTCA CTCGGGCTTCTAAGTACGCTTAGCGCACGGTAGAGCGCAATCCAAGGCTAACCA CCGTGCATCGATGTTGAAAGAGGCCCTCCGTCCTGAATTCTTCAAGTCCCTGGGG TACGGATCCGACG3′.

The 603 sequence is underlined, the TspRI recognition site is in italic, the start of transcription is in bold, the sugar-phosphate backbone cleavage site for Nt.BsmA1 is indicated with ^, and the position of RNAP stalled at +24 position is in bold italic. Details regarding the design of the template and primer sequences will be provided upon request.

NT-SSBs were produced by the incubation of the DNA fragment with Nt.BsmA1 enzyme and the efficiency of the reaction was evaluated by denaturing PAGE. NT-SSBs-containing and intact DNA templates were purified by QIAquick PCR Purification Kit (Qiagen, Hilden, Germany) and assembled into nucleosomes.

### 2.2. Nucleosome Assembly

Nucleosomes were assembled on end-labeled 282-bp DNA templates by histone octamer transfer from chicken erythrocyte donor chromatin lacking linker histones (-H1) at 1:3 DNA-to-chromatin ratio after dialysis from 1M NaCl as described [40,44]. The nucleosomes were analyzed by native PAGE as described [44].

### 2.3. Protein Purifications

Hexahistidine-tagged *E. coli* RNAP was purified as described [49,50]. GreB protein was purified according to published protocols [51].

### 2.4. Transcription

The open complex on the nucleosomal template (40 nM) was formed with *E. coli* RNAP (200 nM) in transcription buffer TB40 (20 mM Tris HCl, pH 7.9, 5 mM MgCl_2_, 40 mM KCl and 1 mM beta-mercaptoethanol; the numerical index corresponds to the concentration of KCl, all reagents are from Sigma-Aldrich, St. Louis, MO, USA) for 7 min at 37 °C. Elongation complexes containing 11-mer RNA (EC−39, where the number indicates the position of the active center of the enzyme relative to the promoter-proximal nucleosomal boundary, was formed by adding 5′-ApUpC, ATP and GTP (20 μM each; all NTPs are from GE HealthCare, Chicago, IL, USA) for 10 min at 21 °C (RT). In the case of experiments with labeled RNA, EC−39 was pulse-labeled in the presence of [α-^32^P]-GTP (3000 Ci/mmol; PerkinElmer, Waltham, MA, USA or Shemyakin and Ovchinnikov Institute of Bioorganic Chemistry, Moscow, Russia), then unlabeled GTP was added to a concentration of 20 μM and reaction mixture was incubated for 5 min at RT. To prevent multiple rounds of transcription initiation, rifampicin was added to a final concentration of 20 μg/mL. In case of nicked template, transcription was resumed by the addition of four NTPs to a final concentration of 200 μM each to EC−39 in TB150 for limited time intervals at RT (see text and figure legends for details). In case ofintact elongation complexes and nucleosome-free DNA, CTP was added to 20 μM in TB40 for 10 min at RT to form EC−5 containing 45-mer RNA. Then, the EC−5 was washed by ice-cold TB40, TB300 and TB150 using Ni-NTA agarose beads (Qiagen, Hilden, Germany), and CTP, UTP and GTP were added to 20 μM each in TB150 for 10 min at RT to form EC+24. Reaction was terminated by the addition of EDTA (to retain intact ECs; Sigma-Aldrich, St. Louis, MO, USA) or phenol (to obtain purified RNA product; Sigma-Aldrich, St. Louis, MO, USA). In case of GreB, the enzyme was added at the final step of transcription to different final concentrations (10, 25 or 50 nM).

### 2.5. DNaseI and Hydroxyl Radical Footprinting

DNase I footprinting was carried out at a 2.5 μg/mL final concentration of labeled templates in the presence of a 10-fold weight excess of unlabeled −H1 chicken erythrocyte chromatin in TB150 (20 mM Tris HCl pH 8.0, 5 mM MgCl_2_, 2 mM β-ME, 150 mM KCl). DNase I (NEB) was added to the final concentration at 20 units/mL for 30 s at 30 °C after formation of the ECs. The reactions were terminated by adding EDTA and KCl to 10 mM and 270 mM, respectively.

Hydroxyl radical treatment of templates was performed as described [52,53]. Conditions were selected to introduce single, randomly positioned NT-SSBs in less than 25% of templates. Level of digestion was controlled by denaturing PAGE.

The samples were resolved in a native gel [54]. Gel fragments containing desired complexes were cut, DNA-purified, and analyzed by denaturing PAGE. The lanes were scanned using OptiQuant 5.0 software (Packard Instrument Co Inc, Meriden, CT, USA) to determine the protected and sensitive regions.

### 2.6. Electron Microscopy and Image Processing

Freshly prepared EC+20with or without NT-SSB were purified using affinity chromatography on Ni-NTA agarose (Qiagen, Hilden, Germany) and applied to the carbon-coated 400 mesh copper grids (Ted Pella Inc., Redding, CA, USA), subjected to glow discharge using the Emitech K100X apparatus (Quorum Technologies, Lewes, UK). Grids were stained for 30 s with a 1% aquatic solution of uranyl acetate and air-dried. Grids were studied in JEOL 2100 electron microscope (JEOL, Japan) under 200 kV accelerated voltage in a low-dose mode using SerialEM software [55]. Images were obtained with magnification 25 k and 40 k using a Gatan Ultrascan 1000 XP charge-coupled-device (CCD) camera (14 μm pixels).

Tomographic studies were performed at ±70° tilt in a JEOL2100 electron microscope (JEOL Japan) at 200 kV accelerated voltage. Tomograms were obtained using IMOD4.9 software (University of Colorado, CO, USA) [56]. The initial 3D reconstitutions were obtained using random conical tilt (RCT) data. For that, ~2500 tilted pairs were picked up using EMAN2.1 [57]. The correction for CTF of the microscope was performed and reference-free class averages were obtained using EMAN2.1. Fifty best classes were used from each sample to obtain the preliminary 3D reconstructions of native complex EC+20 and complex EC+20NT-SSB + 12 with moderate resolution.

To increase the resolution of complex EC+20NT-SSB + 12 ~6000 more single complexes were added to the set to obtain the final 3D structure of EC+20 with the resolution of 15 Å using RELION2.1 [58]. 

The docking of the complex EC+20molecular model (see below) into the obtained 3D density was performed using UCSF Chimera [59] with a cross-correlation coefficient of 0.73.

### 2.7. Modeling the RNAP–Nucleosome Complex EC+20

To model the +20 RNAP-nucleosome complex, the X-ray structure of a transcribing RNA polymerase II complex with complete transcription bubble (PDB ID 5C4X) [60] was first combined with the X-ray structure of a nucleosome core particle (NCP) (PDB ID 3LZ0) [61] by elongating its downstream DNA duplex with a segment of straight B-DNA and connecting it to the corresponding position along the nucleosomal DNA. The DNA linker was connected using the correct DNA base pair step geometry. The chosen length of the straight downstream DNA linker segment effectively controlled the downstream unwrapping of the DNA from the NCP. The Pol II protein part was then replaced by the RNAP structure from *E. coli* (PDB 4JKR) [62] via structure superposition. The modeling pipeline was based on University of California San Francisco Chimera [59] and 3DNA [63] programs. From an ensemble of models with different degrees of downstream DNA unwrapping, a model with the minimal DNA unwrapping but no significant clashes between the atoms of RNAP and NCP was chosen.

To model the closure of the upstream DNA segment onto the NCP the DNase I and hydroxyl radical footprinting data were first analyzed and compared to the known footprinting patterns of NCP [64]. The analysis suggested that the DNA segment at positions −22 to −10 regains contacts with the NCP with the same rotational setting as the last 13 DNA bp seen in the NCP crystal structure. Hence, the location of the upstream DNA at positions −22 to −10 was assumed to match the positions of the last 13 DNA bp in the NCP crystal structure. The straight DNA segment of 21 bp in length was then used to extend DNA downstream of the position −10 up to the position of the DNA nick used in the experiments (positions +12/+13). The end of the straight DNA segment accurately pointed towards the entry site of the transcription bubble. The discrepancy between the end of the straight DNA segment (location of the P atom at position +12 on the non-coding strand) and the next nucleotide in the X-ray structure of the transcription bubble (location of the O3′ atom at position +12 on the non-coding strand) was 25 Å. However, since, in the original Pol II structure positions, +10–+12 were already shown to be unpaired between the DNA strands, we assume that this region is dynamic and flexible enough to accommodate the linkage.

## 3. Results

### 3.1. Experimental Strategy

To study the process of transcription through chromatin, uniquely positioned mononucleosomes formed on the high-affinity DNA sequences were used [65]. Positioned nucleosomes present a polar barrier to transcription by Pol II [66]. Here, 603 nucleosomes in permissive transcriptional orientation were utilized because they could better recapitulate the properties of nucleosomes transcribed in vivo [44]. This experimental system recapitulates the essential characteristics of transcription through chromatin in vivo, e.g., the survival of histones H3/H4 and displacement of histones H2A/H2B during transcription [44,67].

Most of the experiments described below require high amounts of homogeneous complexes that can only be obtained with *E. coli* RNA polymerase (RNAP) that uses the Pol II-specific mechanism of transcription through chromatin in vitro [44,48,66]. Although this experimental system has certain limitations [67], it recapitulates the authentic nucleosome-specific arrest of RNA polymerase on DNA containing SSBs [45]. Therefore, the experiments were conducted using bacterial polymerase as a convenient experimental model.

SSBs positioned at multiple different positions (+12, +22 and +31) induce the efficient, nucleosome-dependent arrest of RNAP with the active center of the enzyme positioned 8–14 bp downstream of the positions of SSBs at physiological ionic strength (150 mM KCl) [45]. The efficiencies of arrest are different for various SSBs, in the following order: +12 ≥ +22 > +31 > +17 [45]. It was proposed that the efficiency of arrest reflects the efficiency of i-loop formation [45]; therefore, the most efficient formation of i-loop involved in the arrest was expected at position +24, after incorporation of SSBs at position +12 in a non-template strand of DNA (NT-SSB) [45,46]. In the majority of the experiments described below, NT-SSB was introduced at the position +12 of nucleosomal DNA to induce RNAP arrest and formation of the i-loop at position +24.

For structural analysis of the elongation complexes that were arrested or stalled at position +24, nucleosomes were assembled to be intact, or contain a +12 single-strand break in non-template strand 603 DNA, and transcribed for a limited time to form elongation complexes that were stalled (templates without the nick) or arrested (with the nick) at position +24 (EC+24) (Figure 1). The position of the active center of the enzyme was mapped using RNA cleavage factor GreB and the samples were analyzed by electron microscopy. In parallel, the same complexes were analyzed by DNase I or hydroxyl radical footprinting.

### 3.2. RNAP Backtracks to Positions +(20–22) after Stalling/Arrest at the Position +24 

The transcription of SSB-containing nucleosomal template by *E. coli* RNAP for 2 min after the formation of EC−39 results in the efficient formation of elongation complex containing the arrested (templates with the nick) or stalled (without the nick) enzyme (Figure 2). The complex primarily contains RNA 74-mer (position +24 bp in the nucleosome).

Since the stalled/arrested enzyme could spontaneously backtrack along DNA, the actual position of the RNAP active center was determined using the GreB transcription factor, as was conducted previously for EC+48 arrested in a nucleosome [68]. GreB stimulates endonuclease activity of *E. coli* RNAP and induces RNA cleavage at the position of the active center (Figure 2A). Incubation in the presence of GreB revealed backtracking of the enzyme by from 2 to 4 nucleotides to positions +(20–22) (Figure 2B). In the presence of SSB in the majority of the complexes, the active center of RNAP is mapped at the position +20 (EC+20), with a small amount of RNAP at positions +22 and +24. Similarly, in the absence of SSB the complexes stalled at the position +24 after transcription in the presence of a subset of NTPs that contain RNAP primarily backtracked to positions +20 and +21. Thus, transcription to the position +24 primarily results in the formation of EC+20.

It was previously shown that elongation complexes arrested in the nucleosome containing SSB+12 remain stable because RNAP can be chased in the presence of all NTPs at 1M KCl when the nucleosomal structure is strongly destabilized [45]. To evaluate the stability of the stalled EC+20 complexes, they were chased in the presence of all NTPs at different concentrations of KCl (Appendix A). RNA can be further extended in the vast majority of complexes, suggesting that stalled EC+20, like the arrested EC+20, remains functionally active (Appendix A).

### 3.3. The i-Loop Is Efficiently Formed during Transcription through the Nucleosome Containing a Nick: Analysis by Footprinting

The analysis of EC+20 stalled in the nucleosome by non-denaturing PAGE (Appendix A) suggests that the complex forms with high efficiency and migrates as a single band during electrophoresis. To analyze the structural features of DNA in the EC+20 complexes containing +12 NT-SSB and intact DNA, the complexes containing end-labeled DNA were incubated in the presence of DNase I or hydroxyl radicals and separated in a non-denaturing gel. DNA was extracted from the bands in the gel corresponding to EC+20 and analyzed in a denaturing gel (Figure 3, Figure 4 and Appendix A).

Dnase I footprinting of DNA in EC+20 that was spontaneously arrested in the nucleosome containing the +12 NT-SSB revealed protection of DNA characteristics for the elongation complex, and the additional protection of both DNA strands behind the enzyme, as compared with the footprint of the nucleosome in the absence of RNAP (Figure 3 and Appendix A for template and non-template DNA strands, respectively). DNA both upstream and downstream of the enzyme in EC+20 is protected from DNase I cleavage (Appendix A), presumably due to its coiling on the surface of the histone octamer after forming an intranucleosomal loop (i-loop) containing the enzyme. The template strand is protected up to –22th position and non-template strand is protected up to –(28–26)th positions according to the data obtained using DNAse I and hydroxyl radical footprinting (Appendix A). Thus, the footprinting data suggest that the i-loop incorporates ~25 nucleotides of the promoter-proximal linker DNA upstream of the enzyme. The downstream region of the template maintains the nucleosomal structure and shows no additional regions of significant sensitivity or protection from DNase I. A minor increase in sensitivity is detectable at the +(35–60) region, most likely due to the presence of a small fraction of the complexes remaining in an open state (Figure 3).

Similar data were obtained using DNA footprinting by hydroxyl radicals (Appendix A). This approach provides a more detailed picture of DNA accessibility and reveals 10-bp periodicity in DNA’s sensitivity to the radicals (Appendix A with alternating blue and red lines), which is characteristic for the nucleosome structure. Thus, the hydroxyl radical footprinting data are consistent with formation of the i-loop after the formation of EC+20 on the nucleosome containing the +12 NT-SSB.

DNA footprinting in EC+20, as stalled in the nucleosome containing intact DNA, shows the less efficient protection of upstream DNA region and promoter-proximal linker DNA (Figure 4 and Appendix A), suggesting that the i-loop forms less efficiently. However, the region of additional protection extends over the similar distance of ~25 nt of linker DNA as compared with the NT-SSB-containing EC+20, indicating that the structure of the i-loop is preserved. The DNA region +(35–60) between RNAP and remaining DNA-histone contacts shows apparent additional sensitivity (Figure 4B, red line), as compared with intact nucleosome and EC+20 containing the NT-SSB, suggesting that nucleosomal DNA is partially uncoiled from the histone octamer. Thus, the data suggest that, in the absence of NT-SSBs, the i-loop is formed less efficiently, but presumably has the same structure as the i-loop in EC+20 containing the NT-SSB (Figure 4B).

Additionally, in the EC+20, there are changes in the footprint of downstream nucleosomal DNA (Figure 4A, black lines near positions +(92–96) and +(105–110)), as compared with the footprint of the intact nucleosome. Changes in the DNase I sensitivity of DNA in these promoter-distal regions suggest that the structure of the nucleosome downstream of the EC is considerably perturbed, perhaps because unconstrained tension is accumulated in front of the transcribing enzyme.

In summary, the i-loop is only efficiently formed in EC+20 in the presence of the NT-SSB at the position +12 and is mostly open behind and partially open in front of RNAP (Figure 4B), presumably allowing for further transcription through the nucleosome.

### 3.4. DNA Is Partially Uncoiled from the Nucleosome in Intact EC+20: Analysis by Electron Microscopy

As a next step towards understanding the structural basis of i-loop formation, elongation complexes EC+20 with and without NT-SSB at position +12 were analyzed using electron tomography with a +/−70 degree tilt. Electron tomography demonstrated that, regardless of the presence of SSB, complexes consist of two interconnected electron densities, one of which can be attributed to the nucleosome (~10 nm size), and one to RNAP (~15 nm size) (Appendix A). Next, we analyzed 2D projection images of intact and NT-SSB+12 complexes. In an intact EC+20 complex, two densities were linked by one thread (Appendix A). However, the EC+20 complex with NT-SSB+12 is more compact, with the two densities positioned in close vicinity (Appendix A). The distance between the centers of mass of RNAP and nucleosome in native EC+20 (without SSB) is ~20 nm, while in EC+20 with NT-SSB, it is 14 nm (*p* < 0.001) (Figure 5B).

Three-dimensional reconstructions of both complexes were performed using the RCT approach because of the apparent preferred orientation of the complexes on the grid. Due to the flexibility of the intact complex, the obtained resolution was moderate. Nevertheless, the reconstruction of the intact complex demonstrated that RNAP and nucleosome are connected by a linker DNA (Figure 5C). The reconstruction of EC+20 complex containing the NT-SSB+12 showed two connectors between RNAP and nucleosome, suggesting that, in this case, the DNA forms an intranucleosomal loop (Figure 5D). 

Taken together with the footprinting data, which show the partial disruption of the i-loop and uncoiling of DNA in front of the enzyme in intact EC+20 (Figure 4A), the obtained EM data suggest that, during transcription, RNAP uncoils DNA from the nucleosome in front of the enzyme, allowing it to progress further unless the intranucleosomal i-loop forms and locks the transcribing enzyme in the loop. These data also suggest that, in intact EC+20 ~17-bp, the DNA region (60Å:36Å = 1.66 DNA helical coils) is more uncoiled from the nucleosome than in EC+20 containing NT-SSB + 12. The data are in good agreement with the observed additional uncoiling of ~20-bp DNA region in intact EC+20, detected by footprinting (Figure 4A).

### 3.5. Modeling the i-Loop

To determine a possible reason for efficient formation of the i-loop at the position +20, various elongation complexes containing RNAP in several closely related positions (from +18 to +25) within the nucleosome were modeled. Only the complex with an RNAP active site at position +20 yielded the reasonable RNAP orientation, which allowed for the upstream and downstream DNA duplexes to align close to the plane of the nucleosomal DNA gyre (Appendix A and Figure 6A). Note that, in the absence of the SSB, DNA upstream of RNAP in EC+20 was not precisely aligned with the nucleosomal DNA gyre and was positioned some distance from the surface of the histone octamer (Figure 6A).

Then, an ensemble of structures with different unwrapping magnitudes of the upstream and downstream DNA segments, as well as different translational and rotational positionings of the upstream DNA fragment, which reestablishes contacts with the nucleosome core particle, were evaluated. Only the following parameters yielded a stereochemically reasonable model consistent with DNA footprinting data. DNA enters the nucleosome core particle at position −22, with the following 13 bp −(22-9) bound at the surface of the histone octamer (Figure 6B). Starting from position −9, the DNA is unwrapped from NCP and a loop is formed. DNA then enters the RNAP active site. The nick on the non-coding strand between positions +12 and +13 creates a hinge that likely allows for the efficient closure of the i-loop behind the transcribing enzyme, and is necessary for the DNA to enter the RNAP binding site at a non-conventional angle (see Figure 6A for a model of the transcription bubble without a nick). The downstream contacts with nucleosome are reestablished at position +47.

To validate the modeled structure of EC+20, we used single-particle EM analysis. We added about 6000 more particles to the set, which allowed for us to solve the 3D structure of the EC+20 complex with NT-SSB+12 (Figure 6A) at a resolution of 15Å (Appendix A). This reconstruction revealed a compact electron density with dimensions ~22 × 15 nm (Figure 6A). This electron density can be divided into two domains: a larger one with dimensions ~15 × 12 nm, which is likely RNAP, and a smaller one with dimensions ~10 × 8 nm, likely the nucleosome. At the obtained resolution, the relative orientation and distance between the nucleosome and RNAP can be discerned. Docking of the obtained atomic model into EM density (Figure 6B) with CC 0.73 showed good agreement with the proposed model (Figure 6A).

In summary, our modeling suggests that the i-loop at position +20 is formed because in EC+20 DNA behind RNAP is best aligned for binding to the open surface of the histone octamer. DNA binding to the histone octamer is greatly facilitated by the presence of NT-SSB at position +12.

### 3.6. Mechanism of i-Loop Formation

The data described above suggest a model for the early stages of transcription through a nucleosome by RNAP II (Figure 7). As RNAP II reaches the nucleosome (Figure 7, complex 1), the transcribing enzyme partially and transiently unwraps DNA from the surface of the histone octamer ahead of RNAP II. Since the elongation complex has a highly asymmetric shape [60], the extent of unwrapping varies depending on the location of RNAP on nucleosomal DNA. In some cases, the unwrapping is minimal, as in EC+42 complex [68], potentially allowing for the formation of very small intranucleosomal DNA loops, such as the Ø-loop [44]. Alternatively, up to 20 bp of nucleosomal DNA can be uncoiled, as is the case in the EC+20 complex (Figure 6A). This more extended DNA uncoiling likely facilitates further transcription through the nucleosome and allows for the formation of a considerably larger intranucleosomal DNA loop (Figure 6B). In this case, the enzyme transcribes through the promoter-proximal part of the nucleosomal template, briefly pauses at the position +24 (Figure 7, complex 2), and backtracks to the position +20 (Figure 2). In the absence of NT-SSBs the upstream DNA region in the majority of ECs remains unwrapped from the octamer, although a transient i-loop is formed on a small fraction of the templates, probably because DNA in the elongation complex is highly bendable to a wide range of angles [69]. The size of the i-loop is too small to allow for the rotation and further progression of RNAP II; therefore, the loop must be open behind the enzyme to allow for further transcription through the nucleosome (Figure 7, complex 3).

The presence of NT-SSB in nucleosomal DNA results in a considerable change in the efficiency of formation of the i-loop. The majority of complexes have a more compact structure with a promoter-proximal DNA region wrapped around the histone octamer (Figure 7, complex 4). In the resulting structure, the angle of DNA exiting the active center of RNAP II is sharper (~30°) than the intrinsic bend in the elongation complex (~130°). Presumably, SSB in upstream areas introduces a hinge (kink) and allows for the bending of DNA and formation of a more stable i-loop. The formation of the i-loop, in turn, induces efficient RNAP arrest.

Taken together with our previous data, the current results suggest that i-loops likely have a size of ~55 bp and are formed with ~10-bp periodicity [45], indicating that they have similar geometries, properties and rotational orientations of RNAP (Figure 7, complex 2). The proximity of RNAP and nucleosome determines the minimal size of the loop.

## 4. Discussion

Our earlier findings revealed a strong RNAP II pausing on templates with SSBs in a non-template strand when DNA is organized in a nucleosome [45]. The arrest sites are localized downstream of the NT-SSBs and accumulate at certain positions along the nucleosomal DNA (+24, +34 and +44 from the nucleosomal boundary). Here, we examined the structural features of the intermediates formed when RNAP is stopped in the +24 position in the presence or absence of the SSB introduced after +12 NPS nucleotides in the non-template strand. We show that, after RNAP reaches position +24, the enzyme can backtrack for ~2–4nt regardless of the presence of the lesion. According to the data obtained using DNA footprinting and electron microscopy, DNA behind the enzyme recoils on the surface of the histone octamer, forming an i-loop that locks RNAP in the arrested state. NT-SSB+12 strongly stabilizes the looped form of the intermediate.

Our data suggest that the i-loop is formed when the active center of the enzyme is positioned +20 nt from the nucleosome boundary. In the presence of NT-SSB+12, RNAP II pauses at position +24 but not at +20; therefore, backtracking is needed to form the stable i-loop intermediate. Is the i-loop structure formed at +24 or +20? It could be proposed that when the enzyme reaches position +24, the DNA-binding surface of the histone octamer behind the enzyme becomes exposed enough to allow for the formation of an i-loop containing the transcribing enzyme within the loop (Figure 7, complex 2). DNA behind the enzyme interacts with the open surface of the histone octamer and forms the i-loop, and then RNAP is backtracked to position +20 to remove the tension. 

The other possible scenario is that RNAP reaches the +24 position and meets the high nucleosomal barrier to transcript the elongation formed by strong DNA-histone contacts in front of the enzyme. According to the DNA footprinting data (Figure 4A, region +(35–60)), the strong contacts affecting RNAP’s progression through the +24 position can be found near +(60–70) region (SHL-1). The region of strong DNA-histone contacts is present at the SHL-1 region [70,71]. Therefore, RNAP pauses near position +24 and backtracking might occur. A similar Pol II backtracking during transcription through the +(20–25) region was previously observed [70]. According to our previous data, similar nucleosomal barriers to the enzyme transcription could also exist at positions +34 and +44 [45]. Molecular modeling suggests that, when RNAP is at position +24, the DNA behind the enzyme is unlikely to form the i-loop. In the experiments with an intact template, the i-loop is detectable on only a small fraction of the ECs (~20%, Appendix A). Therefore, backtracking presumably results in the steric orientation of RNAP, DNA and histone octamers that support the more efficient formation of the i-loop. 

When the enzyme is stopped at the +20 position, the formation of i-loop is still unfavorable for steric reasons (Appendix A). Only in the presence of NT-SSB behind the enzyme (+12) can the DNA bend and bind to the histone octamer, efficiently forming the i-loop. According to our previous data, NT-SSB +17 also promotes RNAP stalling at position +24, but the break at position +22 causes the enzyme to pause at a different position localized further in the nucleosome (at the position +34). Presumably NT-SSB at +17 allows for the DNA to kink, but NT-SSB at +22 is positioned in the active center of the enzyme, preventing DNA kinking and rotation, and thus allowing for further progression of the enzyme. Therefore, the break can act as a swivel if placed in the promoter-proximal part of the i-loop between DNA-histone contacts from +10 to +20 when the active center of the enzyme reaches position +20.

Our data suggest that the i-loops are likely formed or stabilized in the presence of NT-SSBs. Therefore, SSBs in a non-template strand can induce the formation of stable, non-productive transcription intermediate. The inhibitory effect of SSBs on transcription might suggest a possible mechanism for their recognition in vivo through a transcription-dependent mechanism. Accordingly, the in vivo data obtained for different eukaryotic organisms demonstrate the efficient repair of NT-SSBs [22,31,72].

The i-loops with enclosed Pol II can be efficiently formed at different positions along nucleosomal DNA during transcription. This observation suggests that a tendency to maintain a compact state during transcription (and perhaps during other DNA transactions) is a fundamental property of nucleosomes. The formation of compact complexes with a large number of DNA-histone contacts when Pol II is paused in transient inactive complexes likely decreases the probability of losing the histone octamer during transcription, and thus results in more efficient nucleosome survival. The formation of the loops is especially important before position +49 because then the loops could form but are likely to be resolved behind the transcribing complex, resulting in nucleosome recovery and further efficient transcription along the DNA uncoiled from the surface of the histone octamer. The i-loops could be formed more efficiently when transcription is paused due to conditions in the nuclei resulting in a decrease in the rate of transcription, such as a lower local concentration of nucleotides, thus possibly facilitating nucleosome survival. The possible roles of i-loops in transcription and DNA repair are further discussed in [45,46].

Some protein factors that affect the efficiency of transcription through chromatin (FACT, Asf, Spt4/5, etc.) could also affect the stability and formation of i-loops, therefore affecting the efficiency of transcription of damaged DNA and the overall rate of transcription through chromatin. The roles of various transcription factors during transcription through different types of DNA damages are the subject of future studies.

## Figures and Tables

**Figure 1 cells-11-02678-f001:**
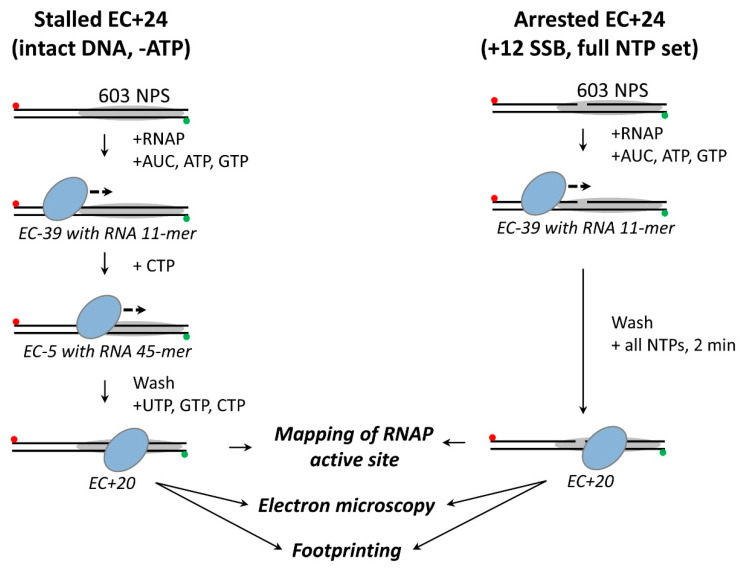
Structural analysis of elongation complexes stalled or arrested in nucleosome. Positioned nucleosomes were assembled on fluorescently labeled 603 DNA containing ROX and fluorescein at the 5′-ends of template and non-template DNA strands (indicated by red and green circles, respectively) containing intact DNA or a single ssDNA break after 12th nucleotide of non-template strand from the nucleosomal boundary, and transcribed for a limited time to form stalled (without the nick) or arrested (with the nick) elongation complex EC+20(the numerical index indicates position of the active site of the enzyme from the nucleosomal boundary). The complexes were treated by DNase I or by hydroxyl radicals, and analyzed by denaturing PAGE. In parallel, the same samples were analyzed by electron microscopy, or position of the active center of the enzyme was mapped using RNA cleavage factor GreB. 603 NPS–603 nucleosome positioning sequence [44,66]. Histone octamer and RNAP are shown by grey and blue ovals, respectively. Direction of transcription is indicated by dashed arrow.

**Figure 2 cells-11-02678-f002:**
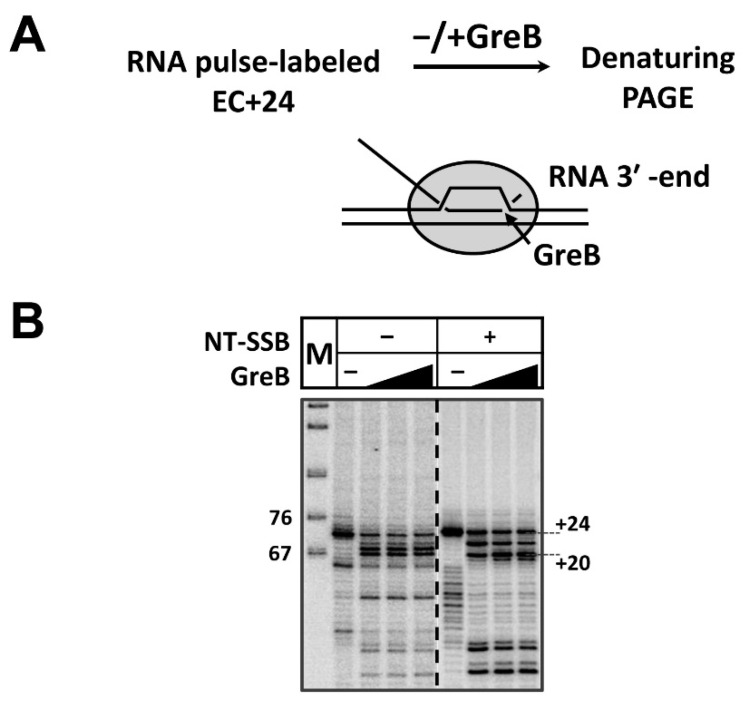
RNA polymerase arrested or stalled at the position +24 in a nucleosome backtracks to form EC+20. (**A**) Experimental approach for mapping positions of the active center of RNAP stalled (without the nick) or arrested (with the nick) during transcription of 603 nucleosome containing NT-SSB after 12th nucleotide. RNA was pulse-labeled and arrested complexes were formed as described in Figure 1. The enzyme can backtrack, disengaging the 3′ end of RNA from the active center. The extent of backtracking was measured using the cleavage factor GreB to stimulate RNA cleavage by RNAP at the active center. (**B**) Analysis of pulse-labeled RNA by denaturing PAGE.

**Figure 3 cells-11-02678-f003:**
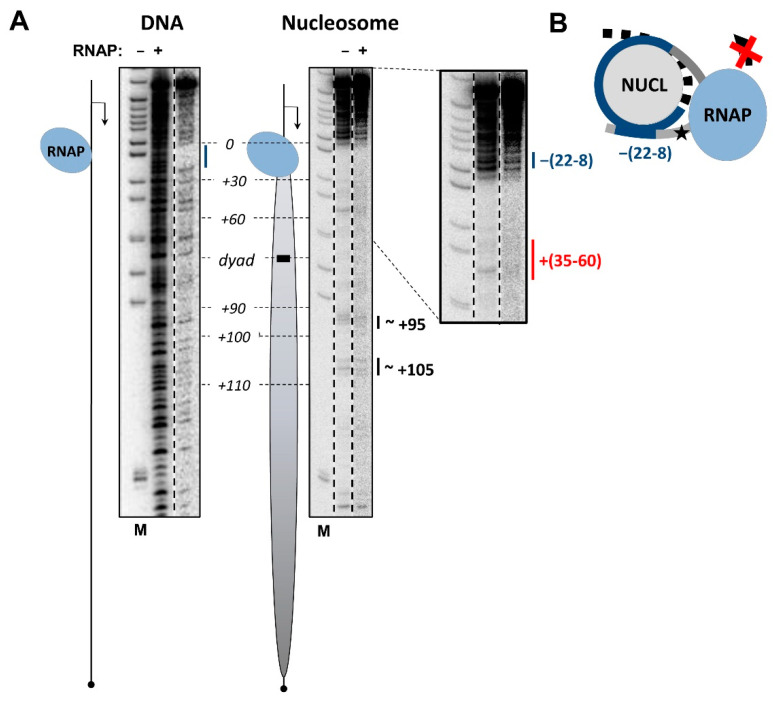
EC+20 containing ssDNA break is flanked by DNA-histone interactions. Analysis of EC+20 structure using DNAse I footprinting. (**A**) EC+20 was formed on the 603 DNA or in the nucleosome containing single ssDNA break after 12th nucleotide (+12 SSB), as described in Figure 1. Analysis of end-labeled DNA (template strand) by denaturing PAGE. Position of the nucleosome on the template is shown by an oval; the direction of transcription is indicated by an arrow. The region protected from DNase I by RNAP on DNA is shown by the blue line. The region of additional protection in EC+20 as compared with intact nucleosome is shown by the blue line. The region of minor additional sensitivity is shown by red line. Note the DNA regions protected from DNase I both upstream and downstream of transcribing RNAP, indicating the formation of an intranucleosomal DNA loop (i-loop). (**B**) Schematic drawing of EC+20 containing SSB: formation of the i-loop. Asterisk indicates the position of +12 SSB on DNA.

**Figure 4 cells-11-02678-f004:**
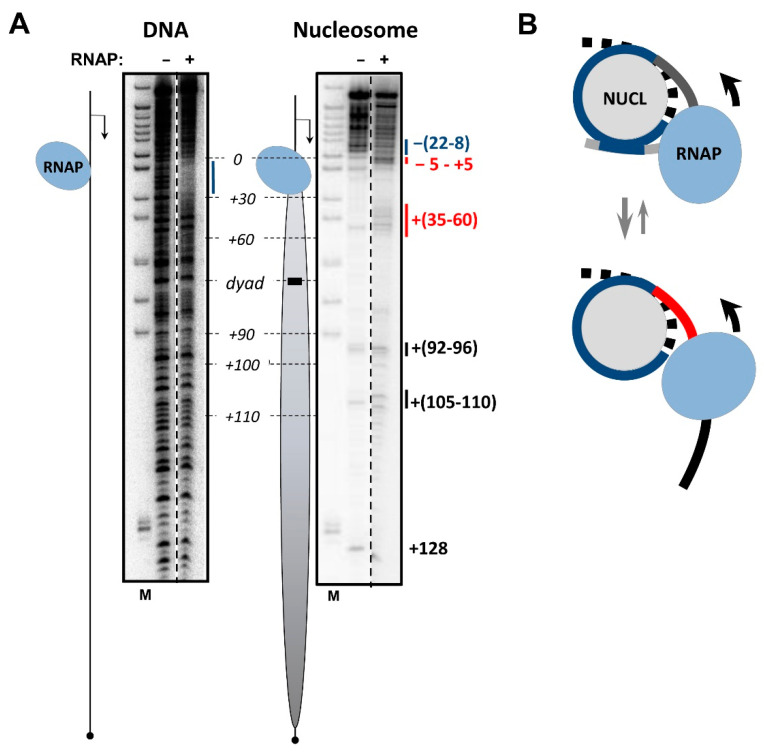
EC+20 containing intact nucleosomal DNA remains mostly in the open state. (**A**) EC+20 were formed on intact DNA or nucleosomes, as described in Figure 1, and analyzed by DNAse I footprinting. Analysis of end-labeled template strand of DNA by denaturing PAGE. Position of the nucleosome on the template is shown by an oval; the direction of transcription is indicated by an arrow. Regions that are either hypersensitive to or protected from DNase I in EC+20 as compared with nucleosomes without EC+20 are shown by red and blue lines, respectively. Changes in the sensitivity of intranucleosomal regions shown by black lines reveal changes in nucleosomal organization downstream the enzyme. (**B**) Schematic drawing of intact EC+20. Linker DNA upstream of RNAP is nearly completely uncoiled from the octamer; the i-loop is only formed on a small fraction of the nucleosomal templates. Region downstream RNAP is partially uncoiled from the histone octamer. Other designations are as in Figure 3.

**Figure 5 cells-11-02678-f005:**
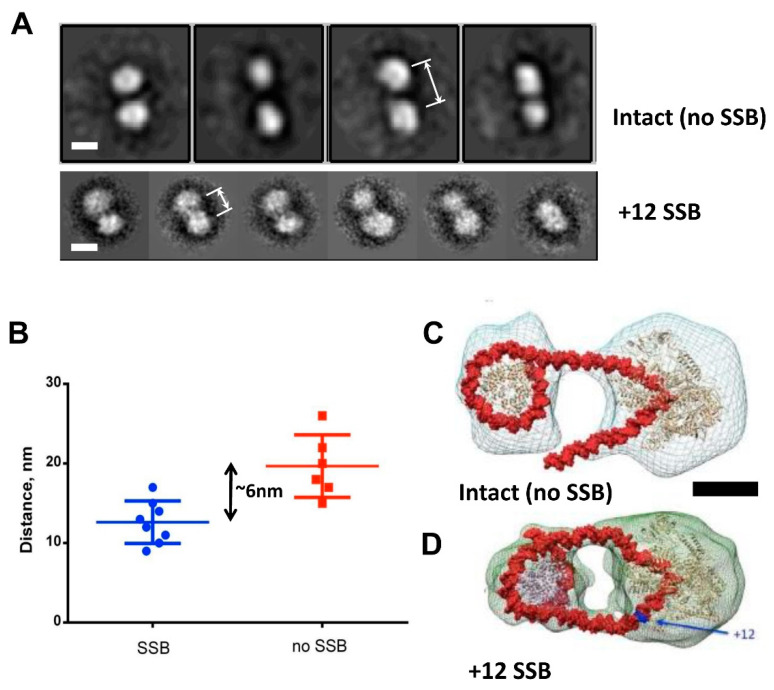
Presence of NT-SSB results in a considerable change in the structure of EC+20. Structure of EC+20 formed on the 603 nucleosomes with and without the NT-SSB was analyzed by electron microscopy. (**A**) Image processing of negatively stained EC+20 – representative class averages, obtained by reference-free classification of particles. Upper row – without SSB; bottom row – with SSB. Scale bars – 10 nm. Arrows indicate approximate average distance between the centers of the particles. (**B**) Distribution of distances between the nucleosome and RNAP in EC+20 with and without the NT-SSB. (**C**,**D**) Preliminary 3D reconstructions of native complex EC+20 (**C**) and complex EC+20 NT-SSB +12 (**D**) with moderate resolution.

**Figure 6 cells-11-02678-f006:**
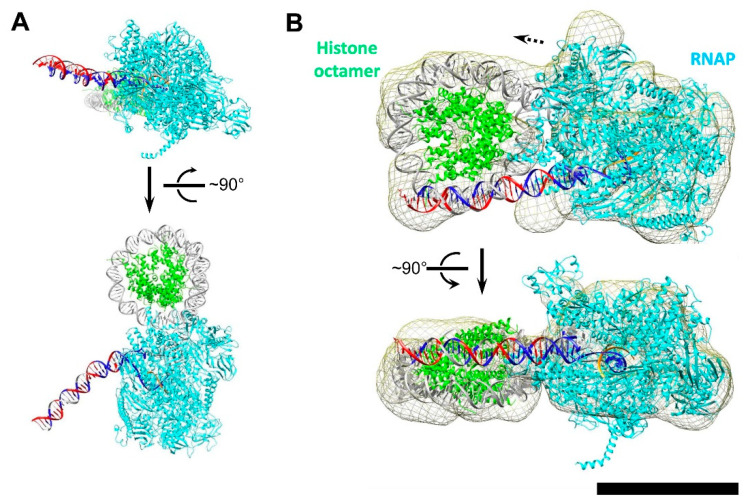
Modeling structure of EC+20 containing ssDNA break into electron density obtained by single particle EM. (**A**) Reconstitution of the EC+20 by fitting structures of nucleosome core particle and *E. coli* RNAP EC (PDB 3LZ0 and 4JKR, respectively). Nucleosomal DNA was connected to DNA localized downstream of the active center of RNAP. The color code is the same as in Figure 4C. Upstream DNA: blue is decreased sensitivity, red is increased sensitivity to DNase I and hydroxyl radicals. The docking of the molecular model of EC+20 complex into 3D density resulted in cross-correlation coefficient 0.73. (**B**) Model of intact EC+20. Color code: RNAP is cyan, histones are green, RNA strand is orange, DNA is grey. Scale bar – 10 nm. Two different orientations of the complexes rotated by ~90 degrees relative to each other are shown.

**Figure 7 cells-11-02678-f007:**
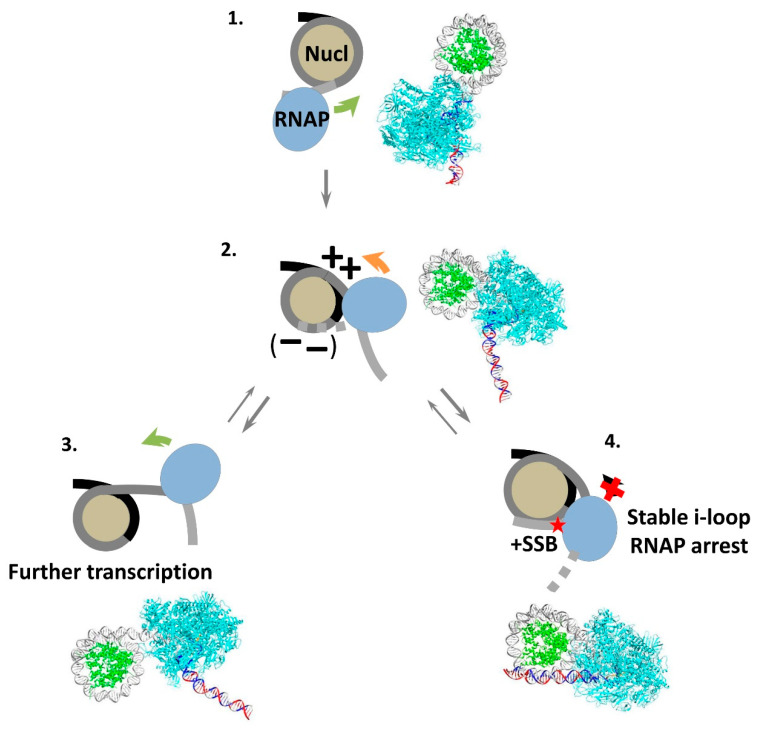
The mechanism of i-loop formation and arrest of RNA polymerase. As RNAP enters a nucleosome (intermediate (**1**)), it partially uncoils nucleosomal DNA from the histone octamer (intermediate (**2**)). Transcription through a nucleosome by Pol II can be accompanied by the formation of transient, small intranucleosomal DNA loops (i-loops) containing the enzyme (intermediate (**2**)) at several locations of the enzyme [45]. Formation of the i-loop could be accompanied by accumulation of negative (--) and positive unconstrained DNA supercoiling (++) behind and in front of the enzyme, respectively. The loops must be opened either in front or behind Pol II to allow for the rotation of the enzyme and further transcription accompanied by nucleosome recovery (intermediates (**2**) and (**3**)). Alternatively, when a single-stranded DNA break is present behind the enzyme in non-template DNA strand, the loop is formed more efficiently, causing the arrest of transcribing Pol II within the i-loop (intermediate (**4**)). Thus, i-loops can serve as sensors of DNA breaks present in non-template DNA strand and hidden in the chromatin structure.

## Data Availability

Not applicable.

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
