# Peer review of "Structure of an Intranucleosomal DNA Loop That Senses DNA Damage during Transcription"

_cells, 2022, doi:10.3390/cells11172678_

Round 1
Reviewer 1 Report
This paper presents a good range of experiments aimed at determining the size and structure of the i-loop formed during RNA polymerase action on nucleosome containing templates , in the presence and absence of single stranded DNA breaks.
I have the following specific comments about the paper
Figure 2 - all the lanes look identical so why is data shown for increasing concentrations of GreB
Figure 3/supp Figure 3 - the differences between +/- RNA pol seems quite subtle especially when the nucleosome is present - it would be useful to signpost the changes much more clearly and also to explain how the figure of 25bp was arrived at for the extra region protected.
How was the loading of the lanes controlled to make sure that in each case equal amounts of template were loaded?
Figure 4 - the difference seem more apparent in this figure.
Figure 5A - why do the shapes of the complexes look so different in the presence and absence of the ss-break. As stated the subunits do seem closer together, but in some cases they also seem to be multi-lobed in the presence of the ssb , rather than just the 2 lobes seen in the absence.
Is it a problem that different nucleotides are present between templates with and without breaks to produce the final complex which is imaged?
In addition to this, although I understand why you would want to use a form of RNA polymerase that is more easily purified it seems that the dimensions of the prokaryotic and eukaryotic polymerases might be quite different as the number of subunits are not the same? In addition the prokaryotic polymerase would not normally encounter nucleosomes and therefore might not behave in exactly the same way as a eukaryotic polymerase which would. It seems that both of these factors might have some effect on the results which are obtained, but this isn’t really discussed anywhere in the paper.
Author Response
Reviewer 1 This paper presents a good range of experiments aimed at determining the size and structure of the i-loop formed during RNA polymerase action on nucleosome containing templates , in the presence and absence of single stranded DNA breaks. I have the following specific comments about the paper Figure 2 - all the lanes look identical so why is data shown for increasing concentrations of GreB
Reply: We have shown multiple digestion points to demonstrate that the observed digestion pattern is at saturation because it does not change considerably after further incubation.
Figure 3/supp Figure 3 - the differences between +/- RNA pol seems quite subtle especially when the nucleosome is present - it would be useful to signpost the changes much more clearly
Reply: To highlight the differences between +/- RNA pol, we have added two new Figures S4 and S5 showing scans of corresponding lanes in Figures 3 and 4. We have also clarified the text of Results (see below). To detect the DNA regions that are protected or hypersensitive to DNaseI or hydroxyl radicals, we compared the scans after aligning with the markers (see new Figures S4 and S5). As DNase I has moderate sequence specificity, the corresponding footprinting patterns characteristic for free DNA, nucleosome and EC+20 can be easily aligned and compared.
and also to explain how the figure of 25bp was arrived at for the extra region protected.
Reply: The following two sentences were added to Results to clarify the measurements: “The template strand is protected up to -22th position and non-template strand is protected up to –(28-26)th position according to the data obtained using DNase I and hydroxyl radical footprinting (Figure S3). Thus, the footprinting data suggests that i-loop incorporates ~25 nucleotides of promoter-proximal linker DNA upstream of the enzyme.”
How was the loading of the lanes controlled to make sure that in each case equal amounts of template were loaded?
For each experiment equal volumes of the same sample were loaded. Each experiment was repeated at least three times. When radioactively labeled DNA was used, the loading was controlled by scintillation counting.
Figure 4 - the difference seem more apparent in this figure.
Reply: This is the expected result due to uncoiling of nucleosomal DNA from the octamer by transcribing RNA polymerase.
Figure 5A - why do the shapes of the complexes look so different in the presence and absence of the ss-break. As stated the subunits do seem closer together, but in some cases they also seem to be multi-lobed in the presence of the ssb , rather than just the 2 lobes seen in the absence.
Reply: We thank reviewer for pointing out this. Indeed, we mistakenly included in Figure 5A raw images of the particles (the original upper row) instead of class-sum averages shown in the bottom row. In the present version we moved single particles of intact complexes to Figure S5C, and inserted class averages for the intact complexes in Figure 5A (upper row). We have also performed a better classification of EC+20 containing the SSB and inserted the updated images in Figure 5A (bottom row). The resulting particles still look somewhat different due to different resolutions of intact (more flexible, lower resolution) and nick-containing complexes.
Is it a problem that different nucleotides are present between templates with and without breaks to produce the final complex which is imaged?
Reply: The sequences of the templates with and without SSB are identical, the only difference is the introduced nick. In addition to this, although I understand why you would want to use a form of RNA polymerase that is more easily purified it seems that the dimensions of the prokaryotic and eukaryotic polymerases might be quite different as the number of subunits are not the same?
In addition the prokaryotic polymerase would not normally encounter nucleosomes and therefore might not behave in exactly the same way as a eukaryotic polymerase which would. It seems that both of these factors might have some effect on the results which are obtained, but this isn’t really discussed anywhere in the paper.
Reply: Indeed, the numbers of subunits for E. coli RNAP and yeast Pol II are quite different. However, the molecular masses (~400 and ~520 KDa, respectively) and structures of E. coli RNAP and yeast Pol II are similar [1,2]. Furthermore, we have shown previously that these enzymes use very similar mechanisms of transcription through nucleosomes [3,4]. The differences between the enzymes have been already addressed in our previous work [5]. In response to the reviewer’s comment, the following sentence was inserted in Results: “Although this experimental system has certain limitations [71], it recapitulates authentic nucleosome-specific arrest of RNA polymerase on DNA containing SSBs [47].”
Reviewer 2 Report
This is a nice contribution from Gerasimova and co-workers. Overall the experimentation is elegant and well performed. Most comclusions are well supported by the experiments.
Major concern:
1. I only have one major concern regarding a far reaching conclusion, which is not supported by biochemical experimentation, and which to me seems too speculative to be included in abstract and overall message of the paper:
The authors finish their Abstract saying: "...i-loop likely plays a role in transcription coupled DNA repair...."
However, this conclusion is drawn on speculation and no real experimental data are presented that directly support such a strong conclusion.
As I understood the results of Figs6 and 7 are theoretical modelling results . Although well done , they suggest that RNAP is stalled at +20 position and that stalling is promoted possibly by recruitment of a region with the SSB. This all goes well together with the nice experimental data presented in Figs 1-5. However, for me it seems to be a rather big stretch to conclude that the function of the i-loop is to arrest RNAP , so that DNA repair can occurr in a transcription dependent manner.
I suggest, either that experiments be performed that really apply DNA damage and transcription repair systems in vitro so that real evidence is supported by biochemical assays. Or alternatively the authors could tone down this conclusion drastically and rather explore this interpretation as a more speculative part in the discussion.
Author Response
Reviewer 2 This is a nice contribution from Gerasimova and co-workers. Overall the experimentation is elegant and well performed. Most comclusions are well supported by the experiments.
Major concern: 1. I only have one major concern regarding a far reaching conclusion, which is not supported by biochemical experimentation, and which to me seems too speculative to be included in abstract and overall message of the paper: The authors finish their Abstract saying: "...i-loop likely plays a role in transcription coupled DNA repair...."
However, this conclusion is drawn on speculation and no real experimental data are presented that directly support such a strong conclusion.
Reply: Corresponding statement in the Abstract has been modified as follows: “Since the i-loop is formed much more efficiently in the presence of SSBs positioned behind the transcribing enzyme, the loop could play a role in transcription-coupled repair of DNA damages hidden in chromatin structure.” The entire text of the manuscript was modified accordingly.
As I understood the results of Figs6 and 7 are theoretical modelling results . Although well done , they suggest that RNAP is stalled at +20 position and that stalling is promoted possibly by recruitment of a region with the SSB. This all goes well together with the nice experimental data presented in Figs 1-5. However, for me it seems to be a rather big stretch to conclude that the function of the i-loop is to arrest RNAP, so that DNA repair can occurr in a transcription dependent manner.
I suggest, either that experiments be performed that really apply DNA damage and transcription repair systems in vitro so that real evidence is supported by biochemical assays. Or alternatively the authors could tone down this conclusion drastically and rather explore this interpretation as a more speculative part in the discussion.
Reply: The connection of the i-loop formation with transcription-coupled DNA repair has been toned down thought the entire manuscript. We have chosen not to include a new paragraph with speculations on this subject to Discussion because we have already discussed this issue previously [6]: no new significant data on this subject has been published since then.
Reviewer 3 Report
Previous studies have suggested the formation of DNA i-loops (intranucleosomal loops) when RNA Pol II transcribes the nucleosomal template. These i-loops, once confirmed, would likely play important roles in regulating chromatin-based transcription, and transcription-coupled DNA damage detection and repair. The current work aims at providing direct evidence for the presence of such i-loops as well as structural insights.
Following the precedence of previous biochemical studies, authors first present good evidence for the preparation and validation of properly assembled complex in vitro using the artificial “603” mono-nucleosome particle and purified bacterial RNA polymerase (for the technical convenience), that are either stalled during transcription (EC+20), with or without a DNA nick engineered in the nucleosome (NT-SSB+12), in which i-loops form at specific locations of the 603 nucleosome.
Structural investigations were then performed using electron tomography and single particle cryo-EM analysis. Here, only structures with low resolution were obtained. At 15-angstrom-resolution (cryo-EM), the relative orientation and distance between the nucleosome and RNAP can be discerned, but not the direct detection of i-loop DNA or other structural specifics. A stereo-permissible structural model is derived from the rough data by docking.
Overall, the work is carefully designed and executed. The presentation is clear and accessible to general readers. However, the low resolution of the EM structures severely limits the conclusions and the mechanistic insights.
Author Response
Reviewer 3 Previous studies have suggested the formation of DNA i-loops (intranucleosomal loops) when RNA Pol II transcribes the nucleosomal template. These i-loops, once confirmed, would likely play important roles in regulating chromatin-based transcription, and transcription-coupled DNA damage detection and repair. The current work aims at providing direct evidence for the presence of such i-loops as well as structural insights.
Following the precedence of previous biochemical studies, authors first present good evidence for the preparation and validation of properly assembled complex in vitro using the artificial “603” mono-nucleosome particle and purified bacterial RNA polymerase (for the technical convenience), that are either stalled during transcription (EC+20), with or without a DNA nick engineered in the nucleosome (NT-SSB+12), in which i-loops form at specific locations of the 603 nucleosome.
Structural investigations were then performed using electron tomography and single particle cryo-EM analysis. Here, only structures with low resolution were obtained. At 15-angstrom-resolution (cryo-EM), the relative orientation and distance between the nucleosome and RNAP can be discerned, but not the direct detection of i-loop DNA or other structural specifics. A stereo-permissible structural model is derived from the rough data by docking.
Overall, the work is carefully designed and executed. The presentation is clear and accessible to general readers. However, the low resolution of the EM structures severely limits the conclusions and the mechanistic insights.
Reply: We agree with the reviewer and added the following sentence to Results: “At the obtained resolution, the relative orientation and distance between the nucleosome and RNAP can be discerned. Docking of the obtained atomic model into EM density (Figure 6B) with CC 0.73 showed good agreement with the proposed model (Figure 6A).”
Round 2
Reviewer 1 Report
I think the authors misunderstood this question ‘ Is it a problem that different nucleotides are present between templates with and without breaks to produce the final complex which is imaged? -probably because I asked about nucleotides when what I meant to ask about was NTPs, as there seem to be different NTPs present during the final reaction . Is it possible for the authors to reply to this question?
Reviewer 2 Report
I think the authors addressed the raised issues in an appropriate fashion. I can now fully recommend acceptance.
Reviewer 3 Report
I am supportive for the publication of the revised manuscript